

# The relationship between leptin and periodontitis: a literature review

Zhijiao Guo[1], Yanhui Peng[1], Qiaoyu Hu[1], Na Liu[2] and Qing Liu[1]

[1] Hebei Key Laboratory of Stomatology, Hebei Clinical Research Center for Oral Diseases, School and Hospital of Stomatology, Hebei Medical University, Shijiazhuang, Hebei Province, China
[2] Department of Preventive Dentistry, School and Hospital of Stomatology, Hebei Medical University, Shijiazhuang, Hebei Province, China

## ABSTRACT

Leptin is a peptide hormone that regulates energy balance, immune inflammatory response, and bone metabolism. Several studies have demonstrated a relationship between leptin and periodontitis, a local inflammatory disease that progressively weakens the supporting structures of the teeth, eventually leading to tooth loss. This article reviews the existing literature and discusses leptin's basic characteristics, its relationship with periodontitis, and its effects on periodontal tissue metabolism.

## INTRODUCTION

Periodontitis is a global disease that is prevalent in both developed and developing countries, affecting approximately 20–50% of the global population. The disease pathogenesis is typically characterized by progressive destruction of the periodontal supporting tissues, which can eventually leading to tooth loss (*França et al., 2017*). As per the Global Burden of Disease (GBD) report (*Chen et al., 2021*), severe periodontitis affects approximately 1.1 billion individuals globally, resulting in a substantial annual public health burden. This underscores the severity of periodontitis as a pressing public health concern. Leptin is a class of adipocytokines discovered in recent years and is involved in metabolism and immunomodulation. Current studies have shown that there is a close relationship between leptin and periodontitis. Furthermore, leptin can influence the pathological process of periodontitis by regulating the immune system, the level of inflammatory factor, bone metabolism, and extracellular matrix of periodontal tissues.

Here, we summarize the relationship between leptin and periodontitis, comprehensively exploring the possible mechanisms by which leptin may affect periodontitis, either positively or negatively, and analyzing potential causal relationships. Clinicians and researchers in the field of dentistry and in the field of endocrinology are provided with references and support.

## SURVEY METHODOLOGY

In this study, three databases (PubMed, China National Knowledge Infrastructure, and Google Scholar) were searched in July 2023 using the following key terms and phrases: (1) "Leptin" as the subject heading or its free terms combined with "periodontitis/periodontal

Corresponding authors
Na Liu, ynlqdx@sina.com
Qing Liu, kqliuqing@sina.com

disease" or their free terms; (2) "Leptin" or its free terms combined with "inflammation/ inflammatory response" or their free terms; and (3) Keywords and subject headings related to "Leptin," "periodontitis" and "immunoreaction/immune response/bone cell/osteocyte/ systemic disease." The inclusion criteria for the articles reviewed were as follows: (1) primary articles; (2) articles published in English or Chinese regardless of study design or publication date; and (3) studies deemed most relevant for this review.

## LITERATURE REVIEW

### Leptin overview

Leptin, a 16 kDa polypeptide hormone, is synthesized from the leptin gene. Adipose tissue is the primary source of leptin (*Zhang et al., 1994*). According to research, other sources of leptin include the salivary gland in the oral cavity and periodontal tissue (*Listl et al., 2015*). The leptin hormone functions through the leptin receptor (LR), a single-transmembrane-domain receptor of the cytokine receptor family. It regulates various biological processes by binding to LRs on cell surfaces (*Shramko et al., 2022*). The Ob-Rb receptor, a long isoform receptor, is considered the most important binding site for transmitting leptin effects. This is because it has the longest intracellular domain, a 30-amino-acid structure, and is the only subtype containing both JAK kinase (Janus kinase) signal sensors and transcriptional activation protein (signal transducer and activator of transcription, STAT) binding sites.

Leptin functions as a hormone as well as a cytokine. As a hormone, leptin acts on hypothalamic appetite-regulating neurons, providing central feedback on the body's energy status. It helps to maintain energy homeostasis and promote energy expenditure by suppressing appetite and regulating body temperature. As a cytokine, leptin has a similar structure to Interleukin-6 (IL-6) and is practically expressed by all immune cells, including neutrophils, monocytes, and lymphocytes (*Yu et al., 2019*). Furthermore, several studies have shown that leptin has potential effects on various inflammatory responses. For example, *Zarkesh-Esfahani et al. (2004)* reported that leptin could induce the expression of IL-6 and Tumor Necrosis factor-$\alpha$ (TNF-$\alpha$) in monocytes cultured *in vitro*, while also activating lymphocytes. *Tian et al. (2002)* reported that leptin can stimulate eosinophil chemotaxis, induce the release of inflammatory cytokines (IL-1$\beta$, IL-6, and IL-8), and monocyte chemoattractant protein-1 (MCP-1) from inflammatory cells. *Mattioli et al. (2009)* found that leptin promoted the differentiation of human Dendritic Cells (DCs), upregulated IL-1$\beta$, IL-6, IL-12, and TNF-$\alpha$ synthesis in DCs, and downregulated IL-10 production in DCs. *Lord et al. (1998)* found that leptin can promote T helper (Th) cell differentiation towards the Th1 subset and regulate effector T cell cytokine secretion, promoting Th1 cell-type cytokine release while suppressing Th2 cell-type cytokine production. Finally, *Agrawal et al. (2011)* concluded that leptin can also activate human peripheral blood B cells, promote IL-6, IL-10, and TNF-$\alpha$ secretion, and induce the phosphorylation of JAK2 and STAT3. Overall, leptin could influence various immune responses by binding to the LRs on the immune cell surfaces.

## The correlation between leptin levels and periodontitis
### The relationship between serum leptin levels and periodontitis

There is a complex bidirectional relationship between serum leptin levels and periodontitis.

Elevated serum leptin levels can exacerbate periodontitis. *Han et al. (2022)* discovered a worsened ligature-induced periodontitis after injecting leptin into the abdominal cavities of mice. Furthermore, high serum leptin levels activate nucleotide-binding oligomerization domain-like receptor protein 3 (NLRP3) inflammasomes, promoting M1 macrophage polarization in a normal body. Activated M1 macrophages can subsequently secrete high amounts of inflammatory cytokines. Additionally, activated NLRP3 inflammasomes can promote the secretion of IL-1β and IL-18 by bone marrow-derived DCs in mice (*Landman et al., 2003*). High serum leptin levels can also downregulate the anti-inflammatory cytokine IL-37 (*Yu et al., 2019*). This chain of inflammatory reactions caused by elevated leptin levels further promotes periodontitis development.

Low serum leptin levels can also impact periodontitis. Lean defective mice, or ob/ob mice, are a mouse model with homozygous mutations in the leptin gene. They cannot produce leptin but still have intact leptin signaling pathways, which enables them to respond to exogenous leptin. *Li et al. (2023)* recently observed the periodontal tissue status and alveolar bone phenotype of ob/ob mice. According to the results, ob/ob mice exhibited higher levels of inflammatory markers and osteoclast markers in periodontal tissue, more distance between the enamel-cementum junction and the alveolar crest, and more severe alveolar bone loss than wild-type mice (*Li et al., 2023*). Additionally, *Hoffmann et al. (2019)* confirmed that the physiological leptin levels (0.1–3.0 mg/kg) can significantly reduce the circulating inflammatory cytokine levels when injected intraperitoneally in ob/ob mice. Low serum leptin levels may thus be a risk factor for periodontitis, whereas low-dose exogenous leptin injections can reduce proinflammatory cytokine levels and promote bone formation.

Periodontitis can also increase serum leptin levels. Serum leptin levels in patients with periodontitis were positively correlated with periodontal clinical parameters in various clinical trials (*Gualillo et al., 2000*; *Karpavicius et al., 2012*; *Lee & Bae, 2016*). However, a recent clinical comparative study demonstrated that patients with periodontitis in high-altitude areas had lower serum leptin levels than a group of 50 healthy controls and that the serum leptin levels were negatively correlated with probing depth and clinical attachment loss (*Yan et al., 2022*). The inverse relationship observed between serum leptin levels and periodontitis in this study may be attributed to the specific sample selection and sample size. The divergent results in the studies discussed earlier suggest that various factors could contribute to alterations in serum leptin levels during the progression of periodontitis. It also highlights that the relationship between periodontitis and serum leptin levels needs further clarification and to be explained by additional relevant molecular mechanisms.

### The role of leptin in the interconnection between periodontitis and systemic diseases

#### Obesity

Obesity is excessive fat accumulation in specific body parts, increasing body weight. Obesity and periodontitis have been positively linked in various studies and analyses of systematic diseases. Obesity and periodontitis have a comorbidity relationship, primarily characterized by shared inflammatory pathways, and they can both cause immune dysregulation (*Ganesan, Vazana & Stuhr, 2021*; *Jepsen, Suvan & Deschner, 2020*).

Adipose tissue is the main source of leptin, and obese individuals secrete more leptin than healthy people. The elevated leptin levels in obese individuals could promote inflammatory cytokine activation in the body, disrupting the oral microbiota and exacerbating periodontitis development (*Chaves et al., 2022*; *Toy et al., 2023*). Furthermore, the obesity-related increase in serum leptin levels lowers the cortical bone density in the alveolar bone area, exacerbating alveolar bone loss. Periodontitis can also affect obesity by influencing leptin. *Li et al. (2015)* discovered that periodontitis could upregulate both leptin and its receptor levels, thus increasing proinflammatory cytokine (IL-6 and IL-8) expression in periodontal ligament cells. Therefore, general periodontitis patients have high levels of various inflammatory mediators (IL-1β, TNF-α, IL-6, and so on) in their periodontal tissues (*Bullon et al., 2009*; *Shimada et al., 2010*). These increased inflammatory mediators could establish a connection between the inflamed periodontal tissues and the rest of the body through the bloodstream. The above-mentioned inflammatory factors can also promote adipose tissue lipolysis, increase liver fat synthesis and lipidation, block hepatic glucose metabolism, and aggravate lipid metabolism disorders (*Fentoglu & Bozkurt, 2008*; *Saito et al., 2008*). Based on the above findings, it is apparent that obesity can promote the progression of periodontitis by upregulating serum leptin levels, while periodontitis can impact the extent of obesity by influencing leptin.

Furthermore, investigating the shifts in the microbial environment in ob/ob obese mice adds a crucial dimension to understanding the link between obesity and periodontitis. On one hand, the absence of leptin can bring about alterations in the gut microbiota. As demonstrated in *Ley et al. (2005)*, ob/ob mice exhibit a significant reduction in *Bacteroidetes* and a corresponding increase in *Firmicutes* in their gut composition. This shift may potentially contribute to the onset of obesity (*Ley et al., 2005*), which in turn can exacerbate the progression of periodontitis. On the other hand, leptin deficiency also leads to transformations in oral microorganisms. In comparison to the wild-type, ob/ob mice demonstrate reduced abundance of two beneficial bacteria, *Akkermansia* and *Ruminococcaceae UCG 014*, that naturally exist in the oral cavity. Additionally, there is an increase in the presence of inflammatory bacteria (*Li et al., 2023*). This suggests that dysbiosis in the oral microbiota significantly contributes to alveolar bone loss in leptin-deficient obese mice.

#### Diabetes

Type 2 diabetes (T2D) can exacerbate periodontitis by influencing leptin. In a past study, serum leptin levels were significantly higher in diabetic patients than in the control group

(*Lai et al., 2020*). Elevated serum leptin levels can enhance the body's inflammatory response and exacerbate periodontitis. Moreover, T2D-induced hyperleptinemia can exacerbate periodontitis *via* aggravating extracellular matrix (ECM) degradation in gingival fibroblasts (*Williams et al., 2016*).

Conversely, periodontitis can impact diabetes by influencing leptin. Periodontitis can upregulate serum leptin levels and, through the JAK2/STAT3 signaling pathway, regulate the involvement of the suppressor of cytokine signaling (SOCS) in cytokine signaling transduction inhibition, which affects T2D incidence (*Zhang et al., 2022*). Moreover, elevated TNF-α in serum worsened diabetes by stimulating C-reactive protein leading to increased insulin resistance and glucose levels (*Anton et al., 2020*; *Santos Tunes, Foss-Freitas & Nogueira-Filho, 2010*). Leptin can also secrete TNF-α by activating blood monocytes (*Constantin & Costache, 2010*). High serum leptin levels in periodontitis suggest elevated severity of insulin resistance.

### Cardiovascular diseases

Serum leptin levels have been shown to be associated with cardiovascular disease. Elevated leptin levels, above 10 ng/L, have been linked to a notable increase in the risk of cardiovascular disease (*Selvarajan et al., 2015*). Moreover, a high concentration of serum leptin can independently serve as a predictive factor for mortality in stroke patients with coronary heart disease (*Puurunen et al., 2017*). Research indicates that leptin plays a role as one of the mediators in the development of atherosclerosis. It has the direct capability to stimulate the recruitment of monocytes to the arterial intima and induce the production of pro-inflammatory cytokines (*Liberale et al., 2017*; *Scheja & Heeren, 2019*), thereby triggering the onset of cardiovascular diseases.

Of note, periodontitis increases the serum leptin levels (*Zhu et al., 2017*), may due to the altered oral microbiome disorder in patients with periodontitis. In patients with periodontitis, the presence of oral *Porphyromonas gingivalis* is significantly higher compared to those without the condition (*Noack et al., 2001*). This bacterium's cell wall product, lipopolysaccharide (LPS), triggers an excessive release of inflammatory factors like TNF-α, IL-6, IL-8, and IL-1β in the oral cavity through the innate immune response, subsequently entering the systemic circulation (*Santos Tunes, Foss-Freitas & Nogueira-Filho, 2010*). Finck's research demonstrated a significant elevation in plasma leptin levels following the intraperitoneal injection of recombinant TNF-α in rats. This indicates that TNF-α induces leptin production *via* the p55 TNF receptor (*Finck & Johnson, 2000*). This implies that the substantial secretion of inflammatory factors in the mouths of periodontitis patients can potentially impact systemic circulation, thus heightening serum leptin levels, posing a threat to cardiovascular health. Furthermore, it should be noted that *Porphyromonas gingivalis* not only indirectly influences leptin levels, posing a threat to cardiovascular health, but the outer membrane vesicles it generates can be released into the environment, promoting the calcification of vascular smooth muscle cells. This calcification is a significant hallmark of atherosclerosis (*Zhang et al., 2021*).

Based on the above discussion, it is apparent that serum leptin levels, systemic diseases, and periodontitis all have a complex and bi-directional relationship. Leptin may be

involved in periodontitis development, whereas periodontitis may affect the pathological process of systemic diseases by influencing leptin. In other words, leptin, to a certain extent, acts as a bridge connecting local tissue inflammation in periodontitis and systemic diseases. Therefore, the dual effects of leptin should be fully considered when treating periodontitis and associated diseases. Oral microbial imbalance can influence both periodontitis and systemic health issues *via* the leptin pathway. Consequently, future research endeavors might emphasize efforts to prevent or restore a balanced oral ecology. This could be achieved through lifestyle modifications or the development of innovative therapies like prebiotics and probiotics (*Păunică et al., 2023*). In addition, a critical focus for upcoming research should involve delving deeper into the systemic regulatory mechanism of leptin and elucidating its association with systemic diseases (*Obradovic et al., 2021*).

## The relationship between local leptin levels in periodontal tissue and periodontitis

The enzyme-linked immunosorbent assay (ELISA) was used to detect the leptin levels in periodontitis patients' (periodontitis patients without systemic diseases) gingival tissue, saliva, and gingival crevicular fluid (GCF). The findings revealed a negative correlation between the degree of periodontitis and leptin levels in the gingival tissue, saliva, and GCF. *Gan et al. (2018)* and *Thanakun, Pornprasertsuk-Damrongsri & Izumi (2017)* reported that leptin levels in gingival tissue decreased as increased severity of periodontitis. In addition, *Purwar et al. (2015)* found that periodontitis patients had significantly lower leptin levels in their saliva than the healthy control group. Lastly, *Johnson & Serio (2001)* discovered that leptin levels in GCF gradually decreased with the aggravation of periodontitis inflammation.

While the local leptin levels in periodontitis patients' periodontal tissue decrease, there is an observed increase in serum leptin concentration. This apparent discrepancy might be explained by the heightened vascular permeability in periodontal tissue during the onset of periodontitis. As the tissue becomes more permeable, it allows for the leakage of leptin from the periodontal tissue, subsequently elevating the serum leptin levels. This phenomenon was highlighted by *Johnson & Serio (2001)*. Furthermore, a study by *Wanichkittikul et al. (2021)* noted a negative correlation between the extent of periodontal tissue damage and local leptin concentration. It suggests that leptin actively participates in repairing periodontal tissue damage. As it binds to local receptors, leptin is consumed in the process, leading to a decrease in its local concentration. This underscores the potential protective role of leptin in maintaining the integrity of periodontal tissue. Some studies have shown that that periodontitis can increase local periodontal leptin levels, which need to be further studied.

## The effects of nonsurgical periodontal treatment on leptin levels

Changes in serum and periodontal tissue's leptin levels following nonsurgical periodontal treatment are critical for determining whether periodontitis and leptin have a bi-directional relationship. A study conducted in Thailand revealed that non-surgical
periodontal treatment can reduce serum leptin and C-reactive protein (CRP) levels in periodontitis patients (*Wanichkittikul et al., 2021*). However, a 2017 meta-analysis found that serum leptin levels in systemically healthy periodontitis patients did not change after periodontal treatment (*Zhu et al., 2017*). The inconsistency in these studies' results may be attributed to the fact that serum leptin levels are affected not only by the condition of periodontal tissues but also by an individual's other body tissues, health status, lifestyle, age, and so on *Soorya et al. (2014)*. However, the underlying mechanisms of these additional factors are unclear and should be clarified further. In patients with periodontitis, leptin levels in GCF increased after root planing alone and root planing with local administration of tetracycline, with the latter having a greater increase, which gradually reverted to a normal level (*Meharwade, Gayathri & Mehta, 2014*). Furthermore, recent clinical studies have demonstrated that non-surgical periodontal treatment can improve clinical parameters (clinical attachment levels, plaque index, probing depth, and so on,) in periodontitis patients and significantly increase leptin levels in GCF (*Ahuja et al., 2019*).

## The effects of leptin on periodontal tissue metabolism
### The effects of leptin on local bone metabolism in periodontal tissue
For adultsthe progression of periodontitis is influenced by the degree of alveolar bone resorption, which is the primary cause of tooth loss (*Sanz et al., 2020*). By acting on the nervous system, leptin regulates local bone metabolism in the periodontal tissue, and it can also directly influence the metabolic state of the alveolar bone.

On the one hand, by acting on the central nervous system (CNS), leptin can indirectly participate in local bone metabolism in the periodontal tissue. Recent research has revealed how leptin influences bone metabolism *via* the CNS. After binding to its receptors on glucose-sensitive neurons in the hypothalamus, leptin activates the adrenal receptor beta 2 (Adrb2) in osteoblasts, downregulates c-myc gene expression, and increases cyclin D production, ultimately lowering osteoblast proliferation. Furthermore, Adrb2 activation promotes the receptor activator of nuclear factorκB ligand (RANKL) expression *via* the protein kinase A and activating transcription factor four pathways, thereby enhancing the function of osteoclasts bone resorption (*Chen & Yang, 2015*; *Reid, Baldock & Cornish, 2018*).

On the other hand, leptin can directly act on the periodontal tissue to regulate bone metabolic processes. Both the human periodontal membrane cells and primary cultured human dental pulp cells have high mineralization ability and certain osteoblast-like characteristics (*Sun et al., 2018*). Leptin has been shown to promote human periodontal membrane cell proliferation, implying that it can promote osteogenesis by influencing osteoblast-like cell proliferation. Following leptin stimulation, the mineralization percentage of bone-forming cells in the periodontal tissue increases, and RANKL mRNA expression decreases, indicating that leptin promotes osteogenesis. Human periodontal membranes are primarily comprised of fibroblasts. When stimulated with leptin *in vitro*, the ratio of osteoprotegerin (OPG) to RANKL in human gingival fibroblasts (HGFs) changes significantly. The OPG/RANKL ratio increases under low-concentration leptin stimulation exerting a bone-protective effect and reducing osteoclast production, but

decreases under high-concentration stimulation (*Guo et al., 2021*). This suggests that varying concentrations of leptin have distinct effects on bone metabolism within the periodontal tissue. Studies have demonstrated that human growth arrest-specific protein 6 (GAS6) is expressed in oral mucosal epithelial cells and stimulate periodontal membrane cell osteogenesis (*Zhang et al., 2020*). In contrast, leptin can reduce the GAS6 levels when stimulating periodontal membrane cells (*Yong et al., 2023*).

In addition to the above-mentioned findings, leptin may promote bone resorption by impacting the expression levels of certain inflammatory factors, including TNF and IL-6, which participate in the activation the function of osteoclast bone resorption (*Faienza et al., 2019*; *Wang & He, 2018*).

Overall, leptin is closely related to alveolar bone metabolism and can influence it in multiple ways. Their relationship is complex, encompassing several signaling pathways, and to fully understand the underlying mechanism, a comprehensive assessment of the interaction is necessary.

## The effects of leptin on extracellular matrix metabolism in periodontal tissue

Excessive degradation of ECM proteins, which leads to inflammatory damage in the periodontal tissue, is an important characteristic of periodontitis. However, there is currently no consensus on how leptin affects ECM.

In an *in vitro* experiment with HGFs, leptin was found to exert a synergistic effect with the proinflammatory cytokine IL-1 and triethylamine salt (pam2CSK4), selectively enhancing Matrix Metalloproteinase 1 (MMP-1) and MMP-3 secretion in HGFs. Excessive protease secretion can accelerate ECM degradation and exacerbate periodontitis. Human periodontal ligament fibroblasts (HPLFs) produce collagen, matrix, elastic fibers, and glycoproteins. High leptin concentrations can also exert a detrimental effect on HPLFs, increasing cell death and accelerating matrix degradation. These results demonstrate that leptin can promote ECM degradation in periodontal tissue. In contrast, when comparing ob/ob mice with wild-type counterparts, it was observed that the gene expression and protein levels of MMP-9 and the proinflammatory cytokine transforming growth factor-β1 (TGF-β1) were notably elevated in the periodontal tissue. This suggests that leptin may confer protection against ECM degradation in periodontal tissues. Nevertheless, further research is warranted to gain a deeper understanding of how leptin affects ECM in periodontal cells.

## CONCLUSIONS

In conclusion, periodontitis is a chronic, highly prevalent, irreversible disease involving inflammation of various oral tissues and characterized by a complex pathogenesis. With the progress of research on leptin, its functions other than regulating body energy metabolism are continuously being discovered. Here, we reviewed current progress on leptin and periodontitis. Leptin has profound effects on periodontitis. It acts as a proinflammatory factor that regulates immunity and is closely linked to alveolar bone metabolism. However, the specific mechanism by which leptin participates in the

pathogenesis of periodontitis, its interaction with inflammatory factors, and its potential immune function in periodontitis remain unclear. Thus, additional research is needed to comprehensively explore leptin's specific characteristics, its relationship with periodontitis, and its effects on periodontal tissue metabolism.

### Funding
The author did not receive funding for this work.

### Competing Interests
The authors declare that they have no competing interests.

### Author Contributions
- Zhijiao Guo conceived and designed the experiments, performed the experiments, analyzed the data, prepared figures and/or tables, authored or reviewed drafts of the article, and approved the final draft.
- Yanhui Peng analyzed the data, prepared figures and/or tables, and approved the final draft.
- Qiaoyu Hu analyzed the data, prepared figures and/or tables, and approved the final draft.
- Na Liu conceived and designed the experiments, authored or reviewed drafts of the article, and approved the final draft.
- Qing Liu conceived and designed the experiments, authored or reviewed drafts of the article, and approved the final draft.

### Data Availability
   This article is a literature review.

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
