# Peer review of "The relationship between leptin and periodontitis: a literature review"

_PeerJ, doi:10.7717/peerj.16633_

## Round 0.1 · original submission · Minor Revisions

This manuscript addresses an interesting topic; however, as pointed out by the reviewers, it requires some technical, contextual and Language revision before being considered for publication. Please address all the points raised by both referees and resubmit for further evaluation. Please remember to view the annotated manuscript attached by Reviewer 2.

**Language Note:** The Academic Editor has identified that the English language must be improved. PeerJ can provide language editing services - please contact us at copyediting@peerj.com for pricing (be sure to provide your manuscript number and title). Alternatively, you should make your own arrangements to improve the language quality and provide details in your response letter. – PeerJ Staff

Reviewer 1 ·

Basic reporting

This article provides a literature review about the links between leptin and periodontitis. By completing an exhaustive literature search on multiple platforms, they discuss key findings that link leptin levels with periodontitis.
This manuscript has potential to provide an adequate literature review for readers to understand the complex relationship between leptin and periodontitis. My comments are relatively minor in part because overall the article addresses several topics well. However, some revisions are necessary. The primary issue that I have with this article is the lack of engagement with the literature on the oral microbiome and its role in regulating periodontitis. For instance, Li et al. (2023, Journal of Periodontal Research) show compelling evidence that leptin-deficient obese mice have different oral microbiomes, specifically downregulating the abundance of particular bacteria. The authors may even want to go beyond the oral microbiome and address how the gut microbiome is impacted by leptin levels which may have an impact on systemic diseases. Mouse models have shown that altering the gut microbiome of leptin-deficient ob/ob mice exhibit a major reduction in Bacteroidetes and a proportional increase in Firmicutes (Ley et al. 2005, PNAS). The ratio between these two taxa have shown to have a role in obesity too.
While the authors have a brief discussion on P. gingivalis, a broader and more detailed discussion on its role with leptin abundance would also strengthen the paper. The should discuss its implications in systemic diseases, particularly cardiovascular disease. They may also want to consider going into more details about how increased serum levels of LPS and TNF-α are associated with P. gingivalis infection which can induce insulin resistance (e.g., Santos et al. 2010, Journal of Canadian Dental Association).

Experimental design

no comment

Validity of the findings

no comment

Additional comments

Minor comments
Authors should consider revising lines 121-122. They should consider, “Lean mice with the homozygous mutation in the leptin gene (ob/ob) are unable to produce leptin even though their leptin signaling pathways are intact.”
They should also consider combining lines 123-124.
In the diabetes section, add more detail. Refer to the Romanian review of the stuff.
How does the microbiome fit in all of this? What sort of studies are the researchers proposing and what steps need to be done in line 198-199.
Italicize Porhyromonas gingivalis on line 190.
Citation needed for line 201-203.
Consider changing ‘mutual relationship’ to ‘bi-directional relationship’.
For line 240-241, the authors should specify that the periodontitis is the primary cause of tooth loss for adults. Caries is more responsible for tooth loss in children.

Reviewer 2 ·

Basic reporting

As the topic has not been reviewed recently the manuscript is of interest. The review is of broad and interdisciplinary interest and within the scope of the journal. The introduction clearly introduces the subject. The text is well-designed with sufficient current references, however the reference mentioned on lines 123 and 158 (Li et al.) is not present in the references section.
Similarly, Zarkesh-Esfahani et al. on line 94, Tian et al.on line 96, Mattioli et al. on line 98, Lord et al. on line 101, Agrawal et al. on line 103, Hoffmann et al.on line 128, Gan et al. and Thanakun et al.0n line 205, Purwar et al. on line 206 do not exist between the references.

Experimental design

Article content is within the Aims and Scope of the journal.As mentioned above, the rferences are not adequately cited. The review is organized logically into coherent paragraphs/subsections.
The term ‘Periodontal Basic Treatment’ on lines 221 and 222 is not an academic statement.The term ‘chronic periodontitis’ is not used currently because of new classification. They must be changed.
The phrase 'According to research' is used several times, but which research is not specified.
The language is clear, intelligible but not professional so it should be edited before accepted.

Validity of the findings

Conclusions are well stated, linked to original research question & limited to supporting results.They also identify the unresolved questions / gaps / future directions.

Additional comments

The article would contribute to the literature. After revisions, it can be accepted and published.Some of the criticized items are highlighted in the attached word file

Annotated reviews are not available for download in order to protect the identity of reviewers who chose to remain anonymous.

---

## Round 0.2 · accepted · Accept

I am delighted to inform you that your paper has been accepted following the successful completion of revisions.

Reviewer 1 ·

Basic reporting

There seems to be several places where there is no space between the citations and the text. For instance, in the tracked changes version, I do not see a space between "group(Purwar...) (L259) and "inflammation(Johnson...)" (L260). I also see the same issue on lines 96,129, 132, 161, These should be addressed before fully accepted.

Experimental design

no comment

Validity of the findings

no comment

Additional comments

The authors have done a nice job in addressing my comments. They address my concerns about their lack of detail about the role of the microbiome in leptin pathways in full detail. If they can just make sure to correct the grammatical errors, the paper is ready for publication.

Reviewer 2 ·

Basic reporting

The authors have revised the manuscript in line with my suggestions in the previous review.

Experimental design

Article content is within the Aims and Scope of the journal. The references are adequately cited. The review is organized logically into coherent paragraphs/subsections.The necessary changes have been done by the authors.A few language errors are highlighted and explanation provided in the attached pdf file.

Validity of the findings

Conclusions are well stated, linked to original research question & limited to supporting results.They also identify the unresolved questions / gaps / future directions.

Additional comments

It can be accepted and published.

Annotated reviews are not available for download in order to protect the identity of reviewers who chose to remain anonymous.